# Neural Combinatorial Optimization with Reinforcement Learning : Solving the Vehicle Routing Problem with Time Windows

## Abstract

In contrast to the classical techniques for solving combinatorial optimization problems, recent advancements in reinforcement learning yield the potential to independently learn heuristics without any human interventions. In this context, the current paper aims to present a complete framework for solving the vehicle routing problem with time windows (VRPTW) relying on neural networks and reinforcement learning. Our approach is mainly based on an attention model (AM) that predicts the near-optimal distribution over different problem instances. To optimize its parameters, this model is trained in a reinforcement learning (RL) environment using a stochastic policy gradient and through a real-time evaluation of the reward quantity to meet the problem business and logical constraints. Using synthetic data, the proposed model outperforms some existing baselines. This performance comparison was on the basis of the solution quality (total tour length) and the computation time (inference time) for small and medium sized samples.

## 1 Introduction

Vehicle routing problem with time windows (VRPTW) can be defined as an extension of the well-known vehicle routing problem (VRP) in which the objective is to design a network of routes to satisfy customers demands with minimal total costsGan et al. (2012). Each route starts from and ends at the depot, and for which the total demand is strictly under the vehicle capacity. Except for the depot, all clients are visited once within the corresponding time window. Moreover, when this previous constraint is violated, a penalty cost will be applied.

Plenty of research have focused on studying and solving this problem commonly referred to as NP-hard Lenstra & Kan (1981). Introducing time windows increases its computational complexity, therefore VRPTW requires more advanced techniques to get reliable solutions. The literature is full of hand-engineered heuristics that provide near-optimal solutions within practical runtime Bräysy & Gendreau (2005). In general, these classical approaches fulfill the trade-off between optimality and complexity. However, the challenge becomes greater when new problem instances are defined or new features are inserted. In this case, a manual adaptation and business knowledge are required to maintain the heuristic's efficiency.

Considering this tedious maintenance process and the huge advancement in learning methods, some works have been focusing on using RNN and RL to learn independent heuristics for solving the VRP. In this paper, we are extending those works by adding the time windows constraint. Precisely, we develop an end-to-end model able to provide near-optimal solutions for every problem instance. In other words, as long as the trained model receives data coming roughly from the same generating process, it yields a reliable solution without the need to build another model for this specific data.

The framework we propose is suitable for introducing more flexibility regarding inputs variance because of the used reinforcement learning approach. The learning process can be formulated as a Markov decision process in a well-defined environment. Therefore, the optimal solution is designed following a dynamic perspective, where the policy is architected through an attention model and trained to maximize the reward which corresponds to the negative tour length. Additionally, the state can be defined as the data relating to each customer (cartesian coordinates, demand, service time, allowed time windows). This state clearly combines static and dynamic parameters. Its dynamic dimension reflects directly the demand change over the learning steps in a manner that once a client

is visited its demand turns into zero . Finally, the environment actions could be seen as the set of customers to include in the solution at each stage.

Our approach is an extension of some existing works namely: Bello et al. (2016), Nazari et al. (2018), and Kool et al. (2018). Our added value lies in generalizing those works to solve the VRPTW, one of the most common combinatorial optimization problems. This paper aims also to strengthen the use of machine learning for solving hard combinatorial problems Bengio et al. (2021). Including time windows constraint increases the VRP complexity and changes the learning and optimization strategy. Consequently, it requires customization in the data generating process, and a new architecture of the reinforcement learning space, especially the environment policy and the transition function. Concretely, the complete model is made up of a neural network which receives the embedding of dynamic and static inputs. The outputs of this sub-model are processed through an attention mechanism within the reinforcement learning space to deliver at the end the near-optimal sequence.

## 2 APPROACH BACKGROUND

Before we deep dive in the model architecture and present its main components, we should briefly highlight some problem-related concepts. In addition to the commonly known ideas about VRP, the treated problem presents some specific assumptions. The halting condition is attained when all nodes demand are satisfied. Furthermore, the vehicle of capacity $D$ returns to the depot to refill when its load runs out without resetting the time. Each customer has a service time $s_i$ strictly lower than the corresponding time window range $[T_{min}^{(i)}, T_{max}^{(i)}]$ to unload its demand. We assume in our case that the time spent to go from customer $i$ to customer $j$ is proportional to the distance $d_{ij}$. In addition, we define $T$ to be the needed time to serve all clients.

In short, VRPTW can be formulated as a graph $G = (V, E)$, such that $X = \{x_0, x_1, ..., x_n\}$ is the set of vertices where $x_0$ stands for the depot, and $E = \{e_{ij} = (x_i, x_j) \mid (x_i, x_j) \in V^2\}$ is the set of edges. Besides, each $x_i$ is associated with a tuple of features $(c_i, d_i, s_i, TW_i)$ and each $e_{ij}$ is associated with a cost $d_{ij}$ , where :

$c_i$: is the two dimensional coordinates of node $i$.

$d_i$: is the demand of node $i$.

$s_i$: is the service time of node $i$.

$TW_i$: is the time window to serve node $i$.

$d_{ij}$: is the distance between node $i$ and node $j$.

The ultimate goal of solving VRP TW is to find the path $\pi = (\pi_1, ..., \pi_N)$ that minimizes the cost within the following space : $\Pi = \{(\pi_1, ..., \pi_N), \pi_i \in \{x_0, x_1, ..., x_n\}\}$.

**Remark:** As a part of the problem setting, it is important to mention that the split deliveries are not allowed, and only one vehicle drive to serve all clients.

Many research in the literature tackled the above-described problem relying on hand-crafted approaches Fisher (1994), Cordeau & Groupe d'études et de recherche en analyse des décisions (Montréal(2000). However, in the recent few years, a new dedicated branch called Neural Combinatorial Optimization has dawned. Precisely, Vinyals et al. (2015) was the first research that presented its main foundations through a sequence to sequence model called pointer network. This latter consists of two coupled RNN, the first one is used to encode inputs to a specific representation, and the other one to decode the processed output and render it as a sequence. In addition, this work follows a supervised learning approach to train this model using labeled data for the traveling salesman problem (TSP). Even though, this study delivered promising insights about using neural networks for solving combinatorial problems, its results are strongly linked to the quality of the used labels. Furthermore it is a hard task to find enough TSP labeled data for training.

To overcome these limitations a reinforcement learning environment to train the RNN models was proposed by Bello et al. (2016). Besides, their approach is fully supported by the fact that roughly all combinatorial problems are evaluated through a specific reward policy. They provide significant results for both TSP and Knapsack problems in terms of the solution quality and the computational time.

Others attempts in the literature considered the same perspective for solving the VRP namely Nazari et al. (2018) and Peng et al. (2019) . Unlike TSP, VRP includes some dynamic parameters, especially regarding node features as the case of the demand which changes after visiting a particular customer. As a consequence, the major changes they proposed deal with the attention mechanism, the transition function in the decoding steps, and the embedding of both dynamic and static inputs.

Taking into account the above-described evolution of neural combinatorial optimization, we will present the model architecture to solve the VRPTW .

## 3 THE MODEL ARCHITECTURE

The configuration encoder-decoder has proved an important efficiency to deal with many problems including VRP Sutskever et al. (2014). As shown in Figure 1, the encoder receives raw data $X$ as described in section 2 and converts it to a convenient representation through many layers. These useful features constructed by the encoder are grasped by the decoder to build progressively the near-optimal sequence. Concretely, the encoder gradually picks out one node to include in the sequence depending on a calculated distribution for each node. Thus, one can give the joint probability of a solution $\pi$ using the chain rule as follows :

$$p_\theta(\pi \mid X) = \prod_{i=1}^{N} p_\theta(\pi_i \mid X, \pi_{1:i-1}) \tag{1}$$

Such that, $\theta$ are the estimated distribution parameters, and $\pi_{1:i-1} = (\pi_1, \pi_2, ..., \pi_{i-1})$.

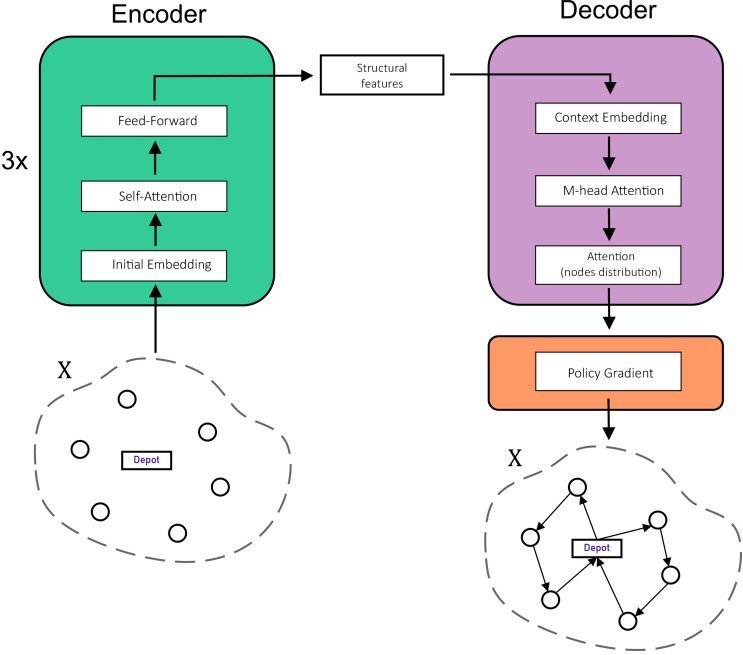

Figure 1: The model architecture Bd-eremeev (2020)

### 3.1 ENCODER

More precisely, the encoder incrementally gets the input sequence and transforms it into a set of embeddings. For each node $x_i$ with features of dimension $d$ ($d = 5$ in the case of VRPTW), the starting embedding $h_i^{(0)}$ with dimension $d'$ is calculated via a linear transformation as follows:

$$\begin{cases} h_i^{(0)} = Wx_i + b & \text{if } i \neq 0 \\ h_i^{(0)} = W_0x_i + b_0 & \text{if } i = 0 \end{cases} \tag{2}$$

Where $W \in \mathbb{R}^{d' \times d}$ and $b \in \mathbb{R}^{d'}$ are the learnable parameters, while $W_0$ and $b_0$ are the ones used for the case of the depot.
As shown in the Figure 1, this set of embeddings crosses a network of 3 layers for further updates. Each layer comprises two sublayers: a self-attention sublayer and then a feed-forward sublayer.

- **Self attention sublayer:** is called also multi-head attention, it brings specific updates to the primary embeddings Vaswani et al. (2017). Let $h_i^{(l)}$ stands for the embedding of the node $i$ in the layer $l \in \{1, 2, 3\}$. Taking into account the recursive relation between layers, one can compute the multi-head attention vector $MHA_i^{(l)}$ following these equations Peng et al. (2019):

$$\left\{ \begin{array}{l} q_{im}^{(l)} = W_m^Q h_i^{(l-1)} \\ k_{im}^{(l)} = W_m^K h_i^{(l-1)} \\ v_{im}^{(l)} = W_m^V h_i^{(l-1)} \end{array} \right\} \tag{3}$$

$$u_{ijm}^{(l)} = q_{im}^{(l)} k_{jm}^{(l)} \tag{4}$$

$$a_{ijm}^{(l)} = \frac{e^{u_{ijm}^{(l)}}}{\sum_y^n e^{u_{iym}^{(l)}}} \tag{5}$$

$$h_{im}^{'(l)} = \sum_{j=0}^{n} a_{ijm}^{(l)} v_{jm}^{(l)} \tag{6}$$

$$MHA_i^{(l)}(h_0^{(l-1)}, h_1^{(l-1)}, ..., h_n^{(l-1)}) = \sum_{m=1}^{M} W_m^O h_{im}^{'(l)} \tag{7}$$

Where, $W_m^Q \in \mathbb{R}^{d \times d'}$, $W_m^K \in \mathbb{R}^{d \times d'}$, $W_m^V \in \mathbb{R}^{d \times d'}$ are learnable parameters, $M$ is the number of heads, $q_{im}^{(l)} \in \mathbb{R}^d$ is the query vector, $k_{im}^{(l)} \in \mathbb{R}^d$ is the key vector, and $v_{im}^{(l)} \in \mathbb{R}^d$ is the value vector.

- **Feed forward sublayer:** Using the multi-head attention vector, the update made up through this sublayer for each node $i$ is computed as follows Peng et al. (2019):

$$\hat{h}_i^{(l)} = tanh(h_i^{(l-1)} + MHA_i^{(l)}(h_0^{(l-1)}, h_1^{(l-1)}, ..., h_n^{(l-1)})) \tag{8}$$

$$FF(\hat{h}_i^{(l)}) = W_1^F ReLu(W_0^F + b_0^F \hat{h}_i^{(l)}) \tag{9}$$

$$h_i^{(l)} = tanh(\hat{h}_i^{(l)} + FF(\hat{h}_i^{(l)})) \tag{10}$$

Such that, $h_i^{(l-1)}$ is the embedding of node $i$ at the layer $l-1$, $W_0^F \in \mathbb{R}^{d \times d'}$, and $W_0^F \in \mathbb{R}^{d' \times d}$.

This computation process is replicated at each layer, consequently the final embedding vector $(h_0^{(N)}, h_1^{(N)}, ..., h_n^{(N)})$ is the one obtained while ending the layer $N$. This output vector is considered as the main structural features for the decoder component.

## 3.2 DECODER

Given the structural features and the constructed solution $\pi_{1:i-1}$, another computation mechanism is applied at each decoding step to bring out a distribution over the next node $i$ to include in the sequence. Firstly, a context vector is computed through the M-head attention mechanism to grant more attention to non-visited nodes and ensure respecting the constraints as well. Explicitly, a concatenation of the embedding inputs is set first through the following operator Peng et al. (2019):

$$h_c' = \left\{ \begin{array}{ll} [\overline{h_t}; h_0^N; D_t; T_t] & \text{if } t = 1 \\ [\overline{h_t}; h_{\pi_{t-1}}^N; D_t; T_t] & \text{if } t > 1 \end{array} \right\} \tag{11}$$

Such that, $[;]$ is the concatenation operator , $\overline{h_t}$ stands for the mean vector of embeddings over all non-visited nodes at the decoding step $t$, $h_{\pi_{t-1}}^N$ denotes the embedding of the node $t-1$ in the partial sequence, $D_t$ is the unconsumed vehicle capacity at the step $t$. Finally, $T_t$ is the accumulated tour time at step $t$.

The context vector $h_c$ is calculated via one M-head attention layer as follows Peng et al. (2019) :

$$\left\{ \begin{array}{l} q_{(c)m} = W_m^Q h_c' \\ k_{jm} = W_m^K h_j^N \\ v_{jm} = W_m^V h_j^N \end{array} \right\} \tag{12}$$

$$u_{(c)jm} = \begin{cases} q_{(c)m}^T k_{jm} & \text{if } d_j \leq D_t, \\ \text{and } x_j \notin \pi_{1:t-1} \text{ and } T_t^j \in TW_j \quad (13) \\ -\infty & \text{otherwise} \end{cases}$$

$$a_{(c)jm} = \frac{e^{u_{(c)jm}}}{\sum_y^n e^{u_{(c)ym}}} \tag{14}$$

$$h'_{(c)m} = \sum_{j=0}^{n} a_{(c)jm} v_{jm} \tag{15}$$

$$h_c = \sum_{m=1}^{M} W_m^O h'_{(c)m} \tag{16}$$

The equation 13 indicates that at each step of decoding, a *masking scheme* Nazari et al. (2018) is applied, which means that the log probability of solution that violates the VRPTW constraints is granted with a value of $-\infty$. In this decoder, we rely on this technique to mask firstly visited nodes, secondly nodes with demand exceeding the remaining vehicle capacity, thirdly nodes with time window not including the quantity $T_t^j$, which is the sum of the expected arrival time and the service time. Besides, the depot can appear many times along the path, for this reason, it is masked only when it is selected at step $t - 1$.

**Remark :** In a step $t$, it is allowed to mask all nodes. This can happen according to the following scenarios :

- If the total vehicle load is over, in this case, it returns to the depot to refill.

- If all nodes are visited, in a such case the vehicle returns to the depot to close the path.

- If the time windows constraint is violated for all non-visited nodes, in this case, the vehicle waits the following time window to continue the decoding process.

Making use of the above equations, one can compute the probability of a potential solution at step $t$ in this way Peng et al. (2019):

$$\begin{cases} q = W^Q h^c \\ k_j = W^K h_j^N \end{cases} \tag{17}$$

$$u_j = \begin{cases} C.tanh(q^T k_j) & \text{if } d_j \leq D_t, \\ \text{and } x_j \notin \pi_{1:t-1} \text{ and } T_t^j \in TW_j \\ -\infty & \text{otherwise} \end{cases} \tag{18}$$

$$p_\theta(\pi_t = x_j \mid X, \pi_{1:t-1}) = \frac{e^{u_j}}{\sum_y^n e^{u_y}} \tag{19}$$

At each decoding step a transition function is run to update the dynamic parameters, namely: the vehicle remaining capacity, the node demand and the accumulated tour time.

Assuming that at the step $t$ the selected node is $j$, therefore:

$$D_{t+1} = D_t - d_j^t \tag{20}$$

$$d_j^{t+1} = 0 \text{ and } d_i^{t+1} = d_i^t \text{ for } i \neq j \tag{21}$$

$$T_{t+1} = T_t + s_j + \alpha d_{ij} \tag{22}$$

Where $\alpha$ denotes the proportionality coefficient between distance and the needed time to reach node $j$ from the previous one $i$. The index $t$ is put on the above parameters to refer to the step $t$.

**Remark:** $s_0$ is the loading time when the vehicle arrives to the depot.

### 3.3 TRAINING

To train the proposed model, we consider the same approach as Peng et al. (2019). Therefore, the stochastic policy $\pi$ is parametrized with $\theta$, and then optimized using a policy gradient. In principal, this latter runs iteratively to estimate the gradient of the constructed solution in order to maximize the reward. This computation process is synthesized in the well-known REINFORCE algorithm Williams (1992).

Considering a space of problems with a specific probability distribution. While training, the instances are built and appended to batches following the same distribution. Using the Monte-Carlo sampling rule the gradients of parameters are defined as follows :

$$\nabla_\theta J(\theta) \approx \frac{1}{B} \sum_{i=1}^{B} (L(\pi_i^s \mid X_i) - L(\pi_i^g \mid X_i)) \nabla_\theta \, log \, p(\pi_i^s \mid X_i) \tag{23}$$

Where $L(. \mid X)$ is the tour length of the solution, $B$ is the batch size, $\pi_i^s$ and $\pi_i^g$ are the solutions of instance $X_i$ constructed by sample rollout and greedy rollout respectively. Explicitly the training algorithm is described through the pseudo-code in algorithm 1.

---

**Algorithm 1** : REINFORCE algorithm

> **Input:** number of epochs $E$, steps per epoch $F$, batch size $B$
> Initialize parameters $\theta$.
> **for** $epoch = 1$ **to** $E$ **do**
>    **for** $step = 1$ **to** $F$ **do**
>       $X_i$=RandomInstance(), $i \in \{1, 2, 3, ..., B\}$
>       $\pi_i^s$=SampleRollout($p_\theta(. \mid X_i)$), $i \in \{1, 2, 3, ..., B\}$
>       $\pi_i^g$=GreedyRollout($p_\theta(. \mid X_i)$), $i \in \{1, 2, 3, ..., B\}$
>       $g_\theta = \frac{1}{B} \sum_{i=1}^{B} (L(\pi_i^s \mid X_i) - L(\pi_i^g \mid X_i)) \times \nabla_\theta \, log \, p(\pi_i^s \mid X_i)$
>       $\theta = Adam(\theta, g_\theta)$
>    **end for**
> **end for**

---

## 4 EXPERIMENTAL RESULTS AND DISCUSSION

To measure the performance of the proposed framework, we conduct three experiments. Each one concerns a particular graph size (VRPTW 10,20,50), and comprises two stages. For the training, we relied on data we generated ourselves, explicitly the model is trained on a batch of 1000 instances and validated on a batch of 500 instances. Similarly, to evaluate the performance, we tested the model on a generated batch of 500 instances. Besides, for The REINFORCE algorithm, we used the Adam optimizer with a learning rate of $10^{-4}$.

In detail, a generated instance consists of a number $n$ (graph size) of nodes. Each node is associated with random coordinates in the space $[0, 1] \times [0, 1]$, also a demand from the discrete set $\{0, 1, ..., 9\}$, then a service time selected randomly from the set $\{0.1, 0.2, 0.5\}$, and finally a time window chosen in the same way from $\{[0, 5], [5, 10], [0, 10]\}$ as we consider the work time divided into three possible service slots. The depot's time window is by definition $[0, 10]$ to enable the vehicle returning to refill at any time, and its demand is always fixed at 0. The vehicle allocated capacity varies depending on the graph size (D = 20 for VRPTW 10, D = 30 for VRPTW 20, D = 40 for VRPTW 50 ).

To compute the accumulated tour time, the coefficient $\alpha$ is thoroughly selected for each graph size to enable serving clients within the total allowed time $[0, 10]$. More precisely, in our case it should satisfy the following inequality: $(\sqrt{2}\alpha + 0.5)n \leq 10$, such that :

$n$ is the corresponding graph size, 0.5 refers to the maximum service time, and $\sqrt{2}$ corresponds to the maximum distance a truck can travel between two points in a square of length 1, and 10 is the time upper bound.

These experiments are run on a GPU machine Tesla K80, using a Tensorflow environment (version 2).

To get clearer ideas about the model performance, a study is led using the same common parameters (node features, coeffcent $\alpha$) to compare between the obtained results and those of other well-known baselines in solving VRPTW problems. On the basis of a test dataset of 500 instances generated

Table 1: Comparison results for VRPTW on the basis of the average tour length on a test dataset.

| METHOD | VRPTW10 | VRPTW20 | VRPTW50 |
|---|---|---|---|
| OR-Tools | 4.75 | 6.26 | 10.78 |
| LKH-3 | 4.84 | 6.37 | 11.03 |
| RNN-RL | 4.78 | 6.21 | 10.96 |

Table 2: Average inference runtime (seconds) for different methods on the test dataset.

| METHOD | VRPTW10 | VRPTW20 | VRPTW50 |
|---|---|---|---|
| OR-Tools | 0.04 | 0.15 | 0.59 |
| LKH-3 | 0.07 | 0.23 | 0.72 |
| RNN-RL | 4.78 | 6.21 | 10.96 |
| RNN-RL | 0.04 | 0.13 | 0.66 |

considering the same distribution as the training one for each graph size, the comparison results are summarized in Table 1. This latter indicates that in terms of the average tour length, the proposed approach clearly outperforms the LKH-3 solver for all graph sizes, and it provides better results as well in comparison to Google's OR-Tools, especially for small sizes.

Regarding the average runtime as shown in Table 2. we can conclude that our method is slightly close to Google's OR-Tools, and doing better than LKH-3 for different sizes.

According to these results, one can view that using this combination of machine learning and reinforcement learning is maintaining good results for different instance sizes. Such performance is mainly due to the greedy decoding procedure, which aims at constructing the solution incrementally and simultaneously reducing the input instance for the next step.

## 5 CONCLUSION

The current work constitutes an extension of the previously mentioned works by including the time windows constraint. Merging neural networks with reinforcement learning proved one more time its efficiency for solving this VRP variant. Even though, we integrate the changes required by a such constraint, especially in the data generating process and in the encoding and decoding procedure the model proves good learning and generalization performance.

The significant results and the flexibility of this method strongly encourage conducting future attempts to include further constraints, particularly multi-depot and heterogeneous fleet and even pick-up and delivery concept.

Considering all these constraints could be time-consuming at the training phase, for this reason, one can incorporate local search notions to build an end-to-end framework for solving all VRP variants.

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

## A  APPENDIX

In this section, we provide a description of the baseline we used for the comparison, namely: LKH-3 and Google's Or-tools. They are well-known and open-source solvers dedicated to VRP variants and other combinatorial optimization problems.

### A.1 GOOGLE OR-TOOLS

Google Optimization Tools (OR-Tools) google (2018) is an open-source solver, commonly known as one among the most performing implementations for solving combinatorial optimization problems. Concretely, it provides an adapted constraint programming interface for solving several mixed-integer linear programming problems. Technically, solving this wide range of variants through OR-tools is based on a mixture of heuristics and metaheuristics, firstly for finding a starting solution (Christofides' heuristic Christofides (1976), Sweep heuristic Wren & Holliday (1972)) and secondly to avoid saddle points while searching for the near-optimal solution (Tabu Search Du & Pardalos (1998) and Simulated Annealing Kirkpatrick et al. (1983)).

### A.2 LK-3

LKH is an open-source solver and a direct implementation of the Lin-Kernighan heuristic Helsgaun (2000) for solving the constrained traveling salesman problem and vehicle routing problems. Over the last decade, many works have been conducted to build on additional features and produce high performing versions. The recent advancements are related to the following concepts:

- Partitioning :
  It paves the way to tackle medium and large-sized problems through partitioning to small instances. Each sub instance is solved individually and its solution is used to improve the global solution.

- Tour merging :
  This concept consists of merging two or even more tours produced for an instance to get the best solution.

- Iterative partial transcription :
  It aims at interchanging parts of two solutions for a given instance problem to get a common best solution. This procedure can be summarized through the algorithm of Möbius et al. (1999).

