# OpenReview forum: "Neural Combinatorial Optimization with Reinforcement Learning : Solving theVehicle Routing Problem with Time Windows"
_ICLR.cc/2022/Conference — ICLR 2022 Submitted_

### Official Review · Reviewer_RKZQ · 2021-10-28

**Correctness:** 1
**Technical Novelty And Significance:** 1
**Empirical Novelty And Significance:** 2
**Recommendation:** 3
**Confidence:** 5

**Main Review:**

The idea of the masking and the multi-head attentions are not novel and they have used earlier on similar problems.

My main concern is the lack of relevant benchmarks. CVRPTW with the size that is considered in the paper can be solved to optimality, see [1]. This exact algorithm can solve small size problems, like yours, in a few minutes. So, you need to provide the exact baseline instead of only comparing it with LKH3. Besides, there are several papers out there which deal with the same problem by RL. For example, [2] follows a similar masking scheme and compare their results with the optimal solution, obtained by Gurobi, on problems of size 20, 50, and 100. A similar problem is studied by [3, 4, 5, 6]. Also, the result of [7], which got online a few days ago is interesting. They obtain the results which are as good as LKH3 on CVRPTW of size up to 300.
Considering the mentioned paper, I do not think the paper at the current shape is ready for acceptance at ICLR. Once you provide larger problems along with the recent benchmarks, the true value of the algorithm will be revealed.


minor comment:
In table 2, there is a wrong row for RNN-RL.


[1] Pessoa, Artur, Ruslan Sadykov, Eduardo Uchoa, and François Vanderbeck. "A generic exact solver for vehicle routing and related problems." Mathematical Programming 183, no. 1 (2020): 483-523.
[2] Zhao, J., Mao, M., Zhao, X., & Zou, J. (2020). A hybrid of deep reinforcement learning and local search for the vehicle routing problems. IEEE Transactions on Intelligent Transportation Systems.
[3] Zhang, K., He, F., Zhang, Z., Lin, X., & Li, M. (2020). Multi-vehicle routing problems with soft time windows: A multi-agent reinforcement learning approach. Transportation Research Part C: Emerging Technologies, 121, 102861.
[4] Lin, B., Ghaddar, B., & Nathwani, J. (2021). Deep Reinforcement Learning for the Electric Vehicle Routing Problem With Time Windows. IEEE Transactions on Intelligent Transportation Systems.
[5] Sultana, N. N., Baniwal, V., Basumatary, A., Mittal, P., Ghosh, S., & Khadilkar, H. (2021). Fast Approximate Solutions using Reinforcement Learning for Dynamic Capacitated Vehicle Routing with Time Windows. arXiv preprint arXiv:2102.12088.
[6] Wang, Y., Sun, S., & Li, W. Hierarchical Reinforcement Learning for Vehicle Routing Problems with Time Windows.
[7] Xin, L., Song, W., Cao, Z., & Zhang, J. (2021). NeuroLKH: Combining Deep Learning Model with Lin-Kernighan-Helsgaun Heuristic for Solving the Traveling Salesman Problem. arXiv preprint arXiv:2110.07983.

**Summary Of The Paper:**

An RL algorithm is proposed to solve CVRPTW with the multi-head attention and masking mechanism. The algorithm is tested on problems with 10, 20, and 50 customers, and is compared with Google OR-Tools and LKH3.


**Summary Of The Review:**

The numerical experiments do not support the claims of the paper and the recent papers and algorithms need to be added in the literature review and the benchmark list.

---

### Official Review · Reviewer_CMcJ · 2021-11-01

**Correctness:** 3
**Technical Novelty And Significance:** 2
**Empirical Novelty And Significance:** 2
**Recommendation:** 3
**Confidence:** 4

**Main Review:**

Strengths
- The goals and approach of the paper are clearly motivated and outlined

Weaknesses
- The writing should be improved overall. Issues throughout include grammar, misuse of commas, capitalization (as in reference to algorithm 1 on page 6), technical details left out, poor formatting.
- There seems to be a lack of novelty in the modeling approach: the same training schemes, DNN model, etc have been developed before, in papers that are cited by this one.
- The results are rather limited and do not seem to show a clear advantage over standard techniques.

Some sentences have unclear meaning, e.g.:
Page 1 - ‘a manual adaptation and business knowledge are needed…”
            - ‘its dynamic dimension reflects directly the demand change over the learning steps…”
Page 2 - ‘Before we deep dive in the model architecture and present its main components, we should briefly highlight some problem-related concepts’   -  the narration is too casual
Page 5 - Equation 13 label is cut off - means that violation is given -infinity in what sense? A large floating point number?
Page 6 - Training:
	       Sample rollout and greedy rollout - are these ever defined? What is the baseline function used in the REINFORCE algorithm?
Page 7 - Google OR-Tools baseline - what are the details of the implementation? Is it the CP-Sat solver? A specialized solver?
            - What is the meaning of the numbers in Table 1? Is lower better?
	    - Why does RNN-RL appear twice in Table 2 with different results?
	    - Is OR-tools called with a solver timeout? Or it is allowed to run to completion?
	    - The results for RNN-RL are very similar to OR-tools. Can you highlight what is thew advantage of your method?

General questions:
- Because the solution is built incrementally, is it possible to take an action that leads to no further feasible actions? (Assuming that feasible solutions require every demand to be met - this isn’t made clear in the VRPTW description). This case is different from those in the remark on page 5 - what is done in this case?
- Generally, does the masking scheme guarantee solutions to be feasible? Is this discussed in the paper?
- What is the novelty of the approach? The network architecture, training scheme, masking, input representation, etc have all been studied before.

**Summary Of The Paper:**

This paper proposes a Neural Combinatorial Optimization approach to solving the Vehicle Routing Problem with Time Windows. It uses a policy gradient method to optimize an attention model, paired with a masking scheme that prevents unwanted actions during the policy rollout. Performance is compared with OR-Tools and LKH-3 solvers.


**Summary Of The Review:**

See strengths and weaknesses described in the Main review.

---

### Official Review · Reviewer_wGHF · 2021-11-02

**Correctness:** 3
**Technical Novelty And Significance:** 1
**Empirical Novelty And Significance:** 1
**Recommendation:** 1
**Confidence:** 4

**Main Review:**

Strengths:
-- The problem of VRPTW is interesting and important

-- The discussion of related work is OK.


Weaknesses:
1) The paper seems to be a purely applied paper. All the techniques used in this paper seem to come from previous works, without many new technical insights. For example,
-- Figure 1 was from Bd-eremeev (2020) WITHOUT a change

-- Equations (3) to (19) are ALL from Peng et al. 2019.

-- The training algorithm (REINFORCE) is not new either

Therefore, it provides little technical contribution to the neural combinatorial optimization literature.

2) The paper does not do a good job in describing the VRPTW, especially

-- what is the main difference between VRPTW and VRP? It is not clearly stated. Perhaps you could first describe the VRP, and then elaborate on the VRPTW and emphasize the TW part.

-- it said that VRPTW brings more complexity and is more challenging, but how exactly is it bringing more challenges? Perhaps the authors could do a complexity analysis to concretely show it. Otherwise, it is very hard to evaluate the difficulty of the new problem and thus it's hard to evaluate the contribution of this paper (from an application point of view).

3) The baselines are weak. The paper claims to be a neural combinatorial optimization, but there are no neural combinatorial optimization methods as baselines?

4) The writing of the paper is quite bad and not polished. For example,

-- Table 2, the 3rd row is a duplicate of the 3rd row of Table 1

-- There should be a space between text and reference

-- The second paragraph of Section 2: E or X -> be consistent

-- VRP TW or VRPTW -> be consistent

-- The reference [Cordeau & Groupe d’´etudes et de recherche en analyse des d´ecisions] is apparently off


**Summary Of The Paper:**

This paper applied the approach in Peng et al. 2019 for vehicle routing problem to a more complex variant called vehicle routing problem with time window.

**Summary Of The Review:**

The paper, in my opinion, is a purely applied paper with almost no change in the method proposed by Peng et al. 2019. Most of the technical contents are copied and pasted from Peng et al. 2019. It's clearly a rejection.

---

### Official Review · Reviewer_kWGX · 2021-11-02

**Correctness:** 3
**Technical Novelty And Significance:** 2
**Empirical Novelty And Significance:** 2
**Recommendation:** 3
**Confidence:** 4

**Details Of Ethics Concerns:**

I do not find any ethical concern with the work.

**Main Review:**

Solving combinatorial problems using machine learning and RL is an interesting and on-going research topic. A few recent works have solve typical OR problems like VRP using RL. This paper extends those ideas to solve VRP problem with additional time windows constraints. The attention based encoder decoder model is a good contribution to model domain constraints for VRPTW problems. However, I have some reservations regarding the novelty of the paper. First, the encoder-decoder model is a simple extension from Peng et. al., 2019. Second, solving VRPTW problem is not entirely novel (as claimed in the paper); e.g., see [1]. Lastly, I think the empirical results are still premature. Although a wide range of large-scale VRP problem instances are available publicly, the authors experimented with a synthetic dataset and compare their approach with simple heuristic methods. To validate the efficiency of the proposed approach, it should be compared with state-of-the-art VRPTW solvers. The presentation of the paper could be improved as well (specially section 3.3 and section 4).

[1] Joe, W. and Lau, H.C., 2020. Deep reinforcement learning approach to solve dynamic vehicle routing problem with stochastic customers. In Proceedings of the International Conference on Automated Planning and Scheduling (Vol. 30, pp. 394-402).

**Summary Of The Paper:**

The paper proposed to solve a vehicle routing problem with time windows using neural network and reinforcement learning framework. An attention based encoder-decoder model is used to predict the distribution over problem instances while satisfying the problem constraints. Then a RL framework is  trained to optimize the model parameter. Experimental results on a synthetic dataset demonstrate that the proposed framework performs at par with traditional combinatorial problem solvers like Google OR tools and LKH heuristic.

**Summary Of The Review:**

The paper shows hope for using ML and RL techniques to solve complex combinatorial problems like VRPTW. The proposed attention based encoder-decoder architecture is a good contribution that shows how to handle domain constraints of VRPTW problems. Having said that I think the paper is still premature in terms of both technical and empirical novelty. The technical contributions are incremental and its performance is not validated against state-of-the-art methods. The empirical evaluation is also not detailed enough to validate the efficacy of proposed method.

---

### Decision · Program_Chairs · 2022-01-20

**Decision:**

Reject

**Comment:**

The reviewers unanimously think the paper has lack of novelty, its contributions are quite limited, and is not ready for publication.